# Physico-Chemical Characteristics of Spodumene Concentrate and Its Thermal Transformations

**DOI:** 10.3390/ma14237423

**Published:** 2021-12-03

**Authors:** Allen Yushark Fosu, Ndue Kanari, Danièle Bartier, Harrison Hodge, James Vaughan, Alexandre Chagnes

**Affiliations:** 1Université de Lorraine, CNRS, GeoRessources, F-54000 Nancy, France; allen.fosu@univ-lorraine.fr (A.Y.F.); ndue.kanari@univ-lorraine.fr (N.K.); daniele.bartier@univ-lorraine.fr (D.B.); 2School of Chemical Engineering, The University of Queensland, Brisbane, QLD 4072, Australia; h.hodge@uq.edu.au (H.H.); james.vaughan@uq.edu.au (J.V.)

**Keywords:** spodumene concentrate, thermal transformation, activation energy, first-order kinetics, morphological changes, physicochemical properties

## Abstract

Spodumene concentrate from the Pilbara region in Western Australia was characterized by X-ray diffraction (XRD), Scanning Electron Microscope Energy Dispersive Spectroscopy (SEM-EDS) and Mineral Liberation Analysis (MLA) to identify and quantify major minerals in the concentrate. Particle diameters ranged from 10 to 200 microns and the degree of liberation of major minerals was found to be more than 90%. The thermal behavior of spodumene and the concentration of its polymorphs were studied by heat treatments in the range of 900 to 1050 °C. All three polymorphs of the mineral (α, γ and β) were identified. Full transformation of the α-phase was achieved at 975 °C and 1000 °C after 240 and 60 min treatments, respectively. SEM images of thermally treated concentrate revealed fracturing of spodumene grains, producing minor cracks initially which became more prominent with increasing temperature. Material disintegration, melting and agglomeration with gangue minerals were also observed at higher temperatures. The metastable γ-phase achieved a peak concentration of 23% after 120 min at 975 °C. We suggest 1050 °C to be the threshold temperature for the process where even a short residence time causes appreciable transformation, however, 1000 °C may be the ideal temperature for processing the concentrate due to the degree of material disintegration and α-phase transformation observed. The application of a first-order kinetic model yields kinetic parameters which fit the experimental data well. The resultant apparent activation energies of 655 and 731 kJ mol^−1^ obtained for α- and γ-decay, respectively, confirm the strong temperature dependence for the spodumene polymorph transformations.

## 1. Introduction

Lithium is undergoing important investigations in order to meet its worldwide stable supply as this element is now considered a critical metal by many countries. It is produced from salar brines or ores. The production from salar brines involves three major steps; evaporation, purification and precipitation. The recovery of the metal by this approach faces many challenges, including the delay in production due to solar evaporation and the presence of many impurities which demand the implementation of refining stages. The main interest of lithium production from salar brine comes from the low cost of the operations. However, the recent increase in the price of lithium has resulted in huge investments into lithium production from ores.

Lithium–Cesium–Tantalum (LCT) pegmatites are the main source of lithium as well as a rich source of tantalum and cesium as they contain spodumene, columbite–tantalite and pollucite, respectively [1]. They may also contain lepidolite, petalite, eucryptite and amblygonite as lithium sources. Muscovite, albite, quartz and feldspar are common minerals that are in close association with LCT; often considered as gangue but may be a by-product of lithium processing. In spite of the high processing cost from ores, the increasing demand and its price make it economically viable. About half of the lithium produced is now obtained from ores [2] and spodumene (LiAlSi_2_O_6_) stands out as the most significant ore-type due to the high lithium content (about 4%) in its pure state as well as its ease of processing.

Spodumene belongs to the clinopyroxene group, occurring naturally as α monoclinic polymorph which is immune to chemical attack due to its compact nature. In this form, lithium and aluminum sit at the *M2* and *M1* octahedral sites respectively, whilst silicon is hosted at the middle of the tetrahedron. The octahedral parts through the sharing of common oxygen atoms, bridge to form chains. The octahedral chains are parallelly arranged to Si in the tetrahedron through O–Si–O ionic bond with the Li and Al atoms. Its original space group is C2/c and in this form, lithium bonds with six atoms of oxygen; with a pair of oxygen atoms having a definite Li–O bond length of 2.120, 2.267 and 2.281 Å [3]. At pressures of around 3.2 GPa, the original stable space group (C2/c) may change to P21/c upon reversible transition [4,5,6] and assumes cell parameters of *a* = 9.471 Å, *b* = 8.400 Å, *c* = 5.223 Å, α = γ = 90.0 degrees and β = 110.2 degrees.

β-spodumene, another polymorph of the mineral consists of two separate tetrahedron structures which are joined in a three-dimensional form with their central parts hosting aluminum or silicon [7]. Lithium in this phase is located between the cavity of five-member rings, formed by the individual tetrahedron. Zeolite-like channels created from the five-member rings run parallel to the (100) or (010) planes [8]. The zeolite-like channels which are parallel to the *a* and *b* axis are comparatively larger and account for the tremendous ion-exchange capacity of β-spodumene during roasting [8]. It has the space group P4_3_2_1_2 with cell parameters *a* = *b* = 7.534 Å and *c* = 9.158 Å [7]. An intermediary metastable γ-phase which transforms to the β-phase upon continuous heating has also been identified. It has a hexagonal symmetry where Si and Al sit at the middle of the tetrahedrons. This polymorph of the mineral starts appearing between 700 °C and 900 °C during the thermal process. 

Lithium is sequentially extracted from spodumene via three processes namely decrepitation, roasting to form soluble salt by sulfation, carbonation, chlorination or fluorination [9]; followed by water leaching of the products of roasting to form an aqueous lithium solution. During decrepitation, the mineral is roasted at high temperatures (above 800 °C) to convert the monoclinic α form to the tetragonal β form. The conversion of α to the β-phase leads to volumetric enlargement of the lattice; making the β-spodumene structure comparatively open for the chemical attack. A series of investigations were conducted to enhance the decrepitation process [3,10,11] using different heating techniques [12] as well as optimizing the extraction efficiency at different leaching conditions [12,13,14,15,16]. 

The Pilbara region in Western Australia has a rich deposit of LCT-spodumene-type pegmatite ore where lithium concentrate is produced and exported. Numerous companies are also on the move to mine this type of ore. The need to understand the physicochemical characteristics of the concentrate prior to beneficiation is crucial for efficient lithium recovery. This study aims to characterize that concentrate as well as understand its thermal transformation behaviors. 

## 2. Materials and Methods

### 2.1. Spodumene Concentrate Preparation

The concentrate was achieved by several unit operations including crushing, grinding, gravity separation (Dense Media Separation (DMS)) and froth flotation. The ore was crushed with primary and secondary crushers to particle size of 32 mm onto a stockpile. It was then fed into High-Pressure Grinding Rolls to reduce the size further to 3.35 mm. This was followed by a two-stage DMS which led to the recovery of course spodumene concentrate. The lower grade unrecovered material was fed into a ball mill which reduced the size to 115 microns prior to tantalite recovery as a by-product via gravity separation. The spodumene-containing fraction was conditioned at a little above 50% pulp density in a neutral pH using sodium oleate as collector. Spodumene in the conditioned material was floated at 30% pulp density and gangue minerals rejected to tailings. This led to 85% recovery and an upgraded concentrate of about 4.6% Li_2_O. The floated material then entered the cleaning stage yielding the concentrate. The concentrate produced as described above was provided by Pilbara Minerals for the study. It consisted of damp, fine-grained material with a significant portion composed of large agglomerated particles that were several centimeters in diameter. The bulk of the material was greenish-brown in color. Breaking apart the agglomerate particles revealed a light grey material. An image of the bulk and grey material is shown in Figure 1. The different coloration observed may be due to the presence of varying concentrations of Fe and Mn which substitute for Al at the various locations. Spodumene is usually associated with Fe which gives it a greenish color. At lower Fe concentration resulting from substitution by Mn, a grey/white color is observed as in this case [17]. 

The as-received sample was formed into a cone and tarp and then cut into approximately two portions. A portion was dried at 60 °C for 48 h after which it was passed through a 3.25 mm screen to break agglomerate particles. The dried material was split into eight subsamples using a rotary splitter. These subsamples were further split using a riffle splitter to generate representative samples of about 1 kg in mass. A representative subsample of the other moist portion was sectioned off by pie splitting which was used to determine the elemental and mineral composition.

### 2.2. Particle Size Distribution

A representative subsample of the dried material was used to determine the particle size distribution. Approximately, 15 g of the sample was slurred in a 1 L stirred baffled reactor containing water. A small amount of detergent was added to enhance breaking the agglomerates and reduce hydrophobicity. This slurry was then sampled using a pipette across the entire depth of the reactor. Sample was transferred to a Malvern Mastersizer for particle size analysis. A total of 5 repeats with two additional rejected runs were completed.

### 2.3. Thermal Treatment

Samples were roasted at varying temperatures from 900 to 1050 °C at 25 °C intervals and residence times of 7.5, 15, 30, 60, 240 and 480 min. A cylindrical quartz crucible was used to hold the sample for heating using a Carbolite Gero electric furnace which heats up to 1500 °C. The furnace provides a constant homogeneous heating 20 cm at its center, therefore, samples placed in the middle have constant temperature throughout the experiment. It was calibrated such that the extent of deviation from the set temperature was ±5 °C throughout the study. The reactor was made of quartz tube placed inside the furnace and the whole setup was operated in air with a flow rate of 25 L/h. The furnace was preheated for 60 min to achieve a steady temperature for samples to attain the reaction temperature within the shortest possible time. This was important for accurate time measurements. Residence time was measured immediately after the sample was introduced into the furnace until the specified time elapsed. Samples were weighed and some portions pulverized with an agate mortar prior to XRD analysis and the other portions kept for SEM analyses.

### 2.4. XRD Analyses

X-ray diffraction (XRD) patterns of samples were collected at ambient temperature, using a CuKα radiation D2 Phaser Bruker diffractometer which is equipped with LYNXEYE detector under 30 kV and 10 mA. The analyses were performed on the powdered samples which have non-oriented flat plates. XRD patterns were recorded between 2θ = 2.5° and 2θ = 70° at a scan step of 2θ = 0.02° and step exposure time of 1 s. Semi-quantitative relative abundances of the phases were estimated using the EVA© software coupled with the PDF2 database of the International Centre for Diffraction Data. The semi-quantitative analyses were performed based on the pattern’s relative heights and I/I_cor_ values by assuming that all crystalline phases were detected and their sum was 100%. 

### 2.5. SEM-EDS Analyses

Two preparation procedures were employed on samples before analysis. First, powdered samples were glued on a graphite support and the surfaces covered with a thin layer of carbon. The second was performed by placing the powdered sample in an epoxy resin and allowed to stand until they hardened. Surfaces of the hardened materials were polished and ultrasonically cleaned with deionized water to achieve a highly polished surface. The surfaces of the samples prepared by the two approaches were coated with a thin film of carbon. The coating is to enhance good electrical conductivity during analysis. 

### 2.6. Mineral Liberation Analysis

Mineral Liberation Analysis (MLA) used in this study makes use of BSE images and point-generated X-ray signals coupled with modern image and pattern identification for its measurements [18,19]. The samples were prepared by allowing the untreated concentrate to stand in resin to harden, followed by surface polishing as in Section 2.5 above. The prepared samples were then subjected to the MLA. Grain particles touching each other due to their settling mode in the resin were detected by DataView with its online program software package and then separated from each other using the shadow/boundary identification procedure. This was to ensure that, all grain particles in the sample were well separated from each other to avoid possible interference which may lead to errors in liberation results and mineral phase identification. Mineral phase identification was achieved by comparing elemental composition X-ray analysis with a standard database. Here, particles of the concentrate (which are in the resin) were divided into their individual mineral grains and their boundaries demarcated using their average BSE grey level. The device performs a systematic point X-ray elemental map which is associated with specific grains based on the differences in the grey level from the BSE map of the composite particles. The X-ray elemental map generated is matched to the corresponding average atomic number (AAN) of each mineral for their identification. In situations where mineral identification uncertainties arise due to overlap of grey level or minerals of similar AAN, the area X-ray analysis detects these anomalies and with the help of X-ray mapping, these minerals are discriminated and identified. Data obtained on the minerals are then stored for presentation in MLA DataView software.

### 2.7. X-ray Fluorescence (XRF) Microscopy and Inductively Coupled Plasma—Optical Emission Spectrometry (ICP-OES)

The bulk assay for the sample was carried out by a commercial laboratory (ALS Environmental Testing, Stafford, QLD) by combining XRF and digestion/ICP-OES for major components and minor components, respectively.

## 3. Results and Discussion

### 3.1. Bulk Chemical Analysis

The result of the elemental composition of the concentrate as oxides obtained from XRF and ICP-OES is presented in Table 1. The lithium concentration was determined as 2.14 wt.%; consequently, the calculated lithium oxide and spodumene concentrations were 4.61 wt.% and 57.39 wt.% respectively. This is in close agreement with the 60.21 wt.% spodumenes indicated by the MLA result (Table 2) and 3 wt.% Li content in spodumene from the Pilbara region reported in the literature [17]. All elements identified are also in agreement with the previous investigation by Aylmore et al. [17] with the exception of Ba and Co which were not identified in this study.

### 3.2. Mineralogy

XRD spectra of the concentrate are shown in Figure 2. Spodumene, mica (muscovite and biotite), quartz, feldspars (orthoclase, albite, anorthite) and the amphiboles are the predominant minerals identified which are typical minerals of spodumene ore of Pilbara. XRD has become the common analytical tool for determining the mineral composition of samples, it has, therefore, been used with the support of other analytical techniques (MLA and EDS) for the finest detection of mineral assemblage in the concentrate.

The modal mineralogy identified by MLA with their concentrations are shown in Table 2. This result is consistent with the XRD analyses, however, spessartine, tantalite, calcite and apatite which appeared as trace minerals were not identified by XRD. An operator decision was made to take into account only minerals that present at least two diffraction peaks (the main and secondary peaks). Thus, minerals with a very low concentration and showing only their main peak, which was difficult to dissociate from the background noise, could have been neglected. This could result in some trace minerals unidentified by XRD contrary to MLA and EDS point analysis. 

The results are in close agreement with the mineralogy of LCT-spodumene ore except with the presence of amphibole which is identified in this work. Elemental assay of the concentrate was also calculated by MLA and the results are presented in Table 3. Computing the oxides of these elements shows a close agreement with chemical analyses in Table 1. The results of this computation are shown in Appendix A.

The particle size of the major minerals was also determined by MLA and the corresponding results are displayed in Figure 3. There are no significant differences between the particle sizes of the minerals in the concentrate. The grain sizes are small with the majority below 100 microns (d80 around 100 microns). At this particle size, the minerals are well liberated as the degree of liberation of all of them is above 90% (Figure 4). Spodumene is the most liberated mineral as its degree of liberation reaches nearly 99%. It follows that the comminution conditions employed for processing this ore are good since they yield concentrate which can be easily processed for maximum lithium extraction, provided the spodumene itself does not passivate. A raw MLA data generated for obtaining Figure 4 is provided in Appendix A.

### 3.3. Morphology, Texture of Particles and Mineralogy

Figure 5 shows the volume distribution and cumulative passing of the concentrate particles. Particle diameter ranges from 10 to 200 microns with about 68% of the total volume being 80 microns in diameter. The d50 and d80 are indicated as 57 and 113 microns respectively. The d80 found is in close agreement with the 100 microns suggested earlier. Considering the higher liberation of spodumene grains, it may be indicated that, d80 of 113 microns and particle diameter of 80 microns are good comminution parameters for processing spodumene ore, particularly pegmatites ores of Pilbara origin.

The BSE of the SEM was further used to investigate the morphology and texture of particles in the concentrate. It was coupled with EDS to determine the elemental composition at some points and hence the mineral composition. Figure 6 and Table 4 are the SEM photomicrograph and spot elemental composition of the concentrate respectively. It is observed (from Figure 6a,b) that, it is a well liberated coarse and loose material with varying particles of several microns in diameter; confirming earlier observation.

Spodumene grains were identified using the Si/Al ratio since EDS cannot detect the presence of lithium. The atomic ratio of Si to Al in spodumene is 2.0 hence all grains with spot elemental composition mainly of Al, Si, O and Si/Al ratio of approximately 2.0 corresponds to spodumene. All spots indicated “1” are identified as spodumene. Figure 6c,d confirm that spodumene is a coarse, dense, crystalline solid with a smooth surface, however, scratches were observed on some grains, which may be due to abrasion during comminution. It suggests that spodumene is a hard mineral that resists wear and tear except when in contact with a harder material. This is confirmed by its relatively high 6.5 to 7.0 value on the Mohs scale of mineral hardness. Spodumene is the major mineral found in the concentrate. It is well liberated but with few mineral associations. Spot “2” has a complex composition with Fe, Mg, Ca, Al, Si, O as the major elements but with varying concentrations at different spots. We attribute these grains to the amphibole group of minerals as indicated in XRD spectra due to the higher concentration of Fe, Mg and Ca at some spots. We could not attribute them to the mica group since the mica group identified by MLA in the concentrate are muscovite and biotite which all contain K in their chemistry but K is absent at these spots. Spots identified as amphibole do not have specific color but ranges from light gray to white depending on the concentration of heavy metals that may be present. The micas (muscovite and biotite) are identified by spot 3 where there is an appreciable higher concentration of K. Spots “4” are composed mainly of Si and O with an O/Si ratio of approximately 2 and therefore identified as quartz. Spots “5” and “6” have a complex composition of Ca, K and Na and are linked to the feldspars, specifically, alkali feldspars and anorthite. We specifically identify spot “5” as albite due to the high concentration of Na at these areas and spots “6” as other feldspars. Hematite was not identified by XRD nor MLA but Spot “7” is identified as such based on the composition. It is composed mainly of Fe and O with an O/Fe ratio of approximately 1.5, confirming its identity. Tantalite and apatite which are typical of LCT-spodumene ore of the Pilbara region are also identified at spot “8” and “9”, respectively. The standard deviations calculated for atomic percent obtained on some mineral grains in the concentrate are shown in Appendix A. Spot “10” is composed of Sn, Ta, N and O. Though the identification of Sn by EDS is a confirmation of elemental composition in Table 1, we could not link the identity of this spot to any mineral since its composition did not match with any. We treat it as an impurity that is locked up in the spodumene grain. All major minerals in the concentrate were identified using spot elemental identification by EDS, confirming XRD and MLA analytical results. Raw data (elemental spectrum with their corresponding intensities, weight percent, etc.) generated by SEM-EDS instrument for mineral phases identified can be found in Appendix A. Other elements such as Cr, Ti, F, Th, Sn, Zn, Cu, Ni, V, and S were also identified which are associated with the fine inclusion of some minerals. Though Ta, Nb, Sn and other valuable metals were observed, their concentrations are too low to be extracted in an economical manner. 

### 3.4. Morphological Changes in Spodumene during Thermal Treatment

The morphological changes in the spodumene grain during decrepitation are of great importance since it gives an indication of the extent of structural changes and openness of the mineral for chemicals to interact with lithium atoms. Changes in the morphology of spodumene grains were studied on the residues treated at 900, 950, 1000 and 1050 °C using the BSE of SEM coupled with EDS. Figure 7 gives the SEM images with spots analyzed by EDS for both polished and unpolished samples. There are cracks and disintegration followed by melting and agglomeration with increasing temperature (Figure 7e–h). At 900 °C, micro-cracks are observed (Figure 7e) which becomes prominent at 950 °C (Figure 7f) such that at 1000 °C (Figure 7g), the spodumene grains had disintegrated and was well open for subsequent processes. From this observation, one may advise that 1000 °C is the ideal decrepitation temperature for processing the concentrate though 1050 °C has normally been used by several researchers for processing spodumene ores. At 1050 °C, melting and agglomeration of spodumene with impurities are observed (Figure 7h). A closer look at a portion in Figure 7h (square “A”) confirms the melting and agglomeration (Figure 7i). We observe several regions of dark and light grey as well as dotted white regions with each coloration corresponding to a mineral phase which are fused together in Figure 7i due to melting. Specifically, spots “1”2”4” and “6” were identified in Figure 7i which corresponds to spodumene, amphibole, quartz and feldspars which were fused together. Tantalite and other minerals were also seen fused with some spodumene at other portions. Most investigations of lithium extraction from spodumene are performed at a decrepitation temperature of 1050 °C, however, we advise 1000 °C as the ideal decrepitation temperature for this concentrate owing to the agglomeration at this temperature which can affect downstream processes and the extraction efficiency of lithium.

Most minerals were identified in the residues and thermal treatment did not have any major influence on their composition except melting and agglomeration of the particles at higher temperatures. A representative elemental composition as determined by EDS for samples treated at 900, 950 and 1000 °C are indicated in Appendix A. Though there was no major effect of thermal treatment on the mineral phases, ionic diffusion was observed in some of the residues at increased temperatures which becomes prominent at 1050 °C. Due to this diffusion, the segregation of the feldspars into their individual minerals as well as differentiating them from the micas by EDS becomes a challenge since they appear to have similar elemental compositions. Table 5 shows some elemental associations with spodumene (spot “1”), amphiboles (spot “2”) and representative composition of micas and feldspars (spot “3”, “5”, “6”) as a result of ionic diffusion between minerals at 1050 °C. The composition of quartz, hematite, tantalite and apatite were not greatly affected regarding the ionic diffusion and are indicated at spots 4, 7, 8 and 9, respectively.

### 3.5. Conversion Extent of α-Spodumene

The extent of conversion of α-spodumene to both γ and β was studied at temperatures ranging from 900 to 1000 °C as a function of residence time (7.5 to 480 min). The result is presented in Figure 8. It is evident that the transformation increases with temperature and residence time. At 900 °C, conversion of the α-phase was observed after 60 min treatment which is characterized by peaks of γ and β in XRD diffractogram in Figure 9a. An almost full conversion was achieved at 975 and 1000 °C as XRD patterns do not show the presence of α-phase after 240 and 60 min, respectively (Appendix A). At 900 °C, transformation increased slightly over time. Only 25% conversion was achieved even after 480 min, 17% of the transformed phase being the β-phase (Figure 8). After 480 min the α-phase was still dominant indicating that the phase transformation is more sensitive to temperature than dwelling time. A sharp increase in conversion is observed by increasing the temperature a little above 900 °C (925 °C). More than 60% transformation was attained after 480 min. Few of the transformed phase (17%) is due to the γ-phase and the majority (50%) are the β-phase. Peaks of the initially dominant α-phase gradually decreased whilst the β-phase increased with residence time (Appendix A), confirming also increasing transformation with time. The higher experimental temperature required a shorter residence time for the transformation and vice versa. Further investigations at 1025 and 1050 °C also did not show any α-phase in the XRD spectrum (Appendix A).

### 3.6. Evolution Extent of β-Spodumene

The formation of β-spodumene is favored by increasing the temperature and residence time (Figure 10). The maximum formed at 900 °C is about 17% while almost 100% was achieved at 1050 °C after 60 min. Temperature is observed to be the most sensitive parameter for the process and once the threshold temperature is attained, a few minutes of heating result in appreciable formation. The temperature of 1050 °C is found as the threshold temperature in this study for treating the concentrate; resulting in almost 100% α-conversion and 85% β-formation in just 7.5 min respectively. This is seen in Figure 9b where only γ and β-phases are present in the diffractogram after 7.5 min of treatment. The threshold temperature for the process was also documented elsewhere [20,21]. Investigations by Peltosari et al. [10] and Salakjani et al. [22] revealed a comparatively higher temperature (1100 ℃) as the threshold for the β-phase formation. We suspect that the difference in temperatures reported in the literature and that presented in this study are due to different gangue concentrations in ores, the heat treatment process as well the nature of calibration applied to the furnace. Peltosari et al. [10] nor Salakjani et al. [22], however, gave detailed information on these parameters in order to confirm this speculation. Additionally, there is an influence due to particle size; a higher reaction rate at a lower particle size was observed by Peltosari et al. [10]. The particle size identified in this concentrate is, however, good enough to achieve considerable mineral liberation as well as a β-phase formation at a comparatively lower temperature.

### 3.7. Evolution of the Relative γ-Spodumene

Peaks of metastable γ-phase are seen in XRD diffractogram from the start of the experiment and persist throughout until after 60 min treatment at 1050 °C where it disappears. This phase might be formed at lower temperatures below the minimum temperature of this study as mentioned by Moore et al. [3] and Peltosaari et al. [10] who identified this phase at 800 and 896 °C respectively. The present observation is in agreement with what was reported at 915 °C by Abdullah et al. [11]. The peaks are not very intense compared to the other phases due to the continuous conversion into β-form as it is formed. Percent evolution of the γ-phase throughout the operating temperatures as a function of the residence time in this study is shown in Appendix A. About 23% was the maximum quantity formed in this study at 975 °C after 120 min (Appendix A), which is comparable to the maximum quantity identified by Abdullah et al. [11] at 1125 °C. Likewise, Moore et al. [3] and Peltosaari et al. [10] recorded 35% and 40% as the highest in their studies, respectively. According to Abdullah et al. [11] and Gasalla et al. [23], the quantity of this phase formed is influenced by the particle size of the feed as well as the heat treatment technique employed; finer particles impact amorphicity which easily recrystallizes into the γ-phase on heating compared to larger-sized particles. The heating rate employed also influences the quantity formed; slow heating rates form higher amounts due to the slow rate of conversion to the β-form. There was no major change in the quantity of this phase formed as a function of the residence time in the present work. This may be as a result of the constant heating rate which was maintained throughout the study, hence converting it to the β-form at the same rate. Though Moore et al. [3] indicated the importance of residence time on the transformation, they observed only a marginal 20% increase in this phase after increasing time from 45 min to 240 min at 981 °C. 

It is interesting to note that, the shape of the curves in Appendix A may predict the rate of its formation and conversion into the β-phase. There is a gentle rise and fall of the curve at lower residence time which becomes steeper with increasing residence time. This indicated a lower rate of formation and conversion at lower residence time and vice versa. From Figure 11, both γ- and β-phases evolve from the onset of the experiment but their concentration varies considerably; the quantity of γ-phase being lower than the β-phase at all times. This observation is contrary to research by Moore et al. [3] who indicated that both phases occur in equal amounts with no preference of formation of one phase over the other. We also find that the quantity of each phase formed is temperature dependent with increasing temperature favoring the quantities evolved; which is also in opposition to their findings. At 975 °C, almost all the α-phase had decayed and subsequent β-phase formation was solely dependent on the available γ-phase. Dessemond et al. [24] investigated the effect of this phase concomitant with the β-phase and its effect on lithium extraction. They indicated that its presence adversely affects the lithium extraction if its content is above 10%, however, below it, its effect is minimal, and a typical industrial lithium recovery of 95% is attainable. This therefore calls for a closer look at the ideal temperature and residence time required for the porcess, paying attention to the economics, possible melting and agglomeration and its effects on downstream processes.

### 3.8. Phase Transformation and Kinetic Parameters

The phase transformation of spodumene during thermal treatment was extensively studied by many researchers [3,10,11] revealing α, β and γ as the phases which are present during the process. The transition is generally known to follow the following pathways:(1)$$\alpha \;\overset{k_{1}}{\to }\;\gamma +\beta$$
(2)$$\gamma \;\overset{k_{2}}{\to }\;\beta$$ where $k_{1}$ and $k_{2}$ are the rate constants for α and γ decay, respectively.

Thus, from Equation (1), α-spodumene decays to form γ and β phases. The γ-phase which is formed finally transforms to the β-phase (Equation (2)). Previous studies have suggested a first-order kinetic model for the two transitions. We therefore apply this kinetic theory to both transitions to estimate the fitting kinetic parameters. From an integrated rate Equations of (1) and (2), it follows that:(3)$$\ln\;\alpha_{t}=\ln\;\alpha_{o}-k_{1}t$$
(4)$$\ln\;\gamma_{t}=\ln\;\gamma_{o}-k_{2}t$$
where $\alpha_{o}$ and $\gamma_{o}$ are the initial concentration of α- and γ-phase; $\alpha_{t}$ and $\gamma_{t}$ are concentrations of α- and γ-phase at time t, respectively.

We estimate the rate constants for the decay processes ($k_{1}$ and $k_{2}$) by plotting data in Appendix A using Equations (3) and (4). The resulting $k_{1}$ and $k_{2}$ are indicated in Table 6. These values are found to increase with increasing temperature and they confirm the sensitivity of the processes to temperature. The activation energy can be obtained from a linearized Arrhenius Equation (5):(5)$$\ln k=\ln A-\frac{E_{a}}{RT}$$
where *k* is the rate constant; *A*, the pre-exponential factor; *Ea*, the activation energy; *R*, the gas constant; *T*, the absolute temperature. The apparent activation energies obtained from the Arrhenius plots (Figure 12) for both decay processes using our data in Appendix A are 655 and 731 kJ mol^−1^, respectively. Thus, a comparatively larger amount of energy is required for the decay of γ-spodumene in the second phase of the process than the decay of the α-phase. This is expected due to the higher temperature required for the decay of α-phase prior to the γ-decay. However, α-decay occurs at a faster rate compared to γ-decay (Table 6). Estimated standard errors) for the regression for obtaining the activation energies are very minimal (0.1297 and 0.0489 respectively for α and γ decay); suggesting a good fit and reliable activation energy values for a first-order kinetic model. We record lower apparent activation energy for α-decay than reported in the literature [3]. Abdullah et al. [11] suggested that γ-phase may evolve either from amorphous or crystalline spodumene and suggested the following reaction pathways for the process;
Amorphous spodumene⟶γ-spodumene⟶β-spodumene(6)
Crystalline α-spodumene⟶γ-spodumene⟶β-spodumene(7)

The path of the mechanism may influence the overall activation energy. They predicted low and high activation energies for (5) and (6), respectively. They indicated that finer particles introduce amorphicity, which easily crystalizes out with reduced activation energy. Comparing their particle size to what was used in the current study reveals that our sample is finer and more amorphous than theirs. This suggests that the evolution of the γ-phase from α-spodumene in this study is from the amorphous phase rather than crystalline. Reaction path (5) is therefore suggested for this study and as a result, contributes to the lower activation energy recorded for α-decay in this work than in previous studies.

## 4. Conclusions

The physicochemical characteristics of spodumene concentrate from the Pilbara region were studied. The d50 and d80 of the sample were obtained to be 57 microns and 113 microns, respectively. Particle diameter ranges from 10 to 200 microns with about 68% of the total volume being 80 microns in diameter. At these conditions, spodumene is well liberated at approximately 99%, indicating that sufficient particle size reduction was achieved to enable further processing. Mineral phase identification by the analytical techniques employed in this study is consistent with each other for major minerals. The phase transformation of spodumene from 900 to 1050 °C and 7.5 to 480 min dwelling time was studied using XRD. α transformation was complete at 975 and 1000 °C whilst that of γ occured at 1025 °C and 1050 °C all at different treatment times. About 23% was the maximum quantity of γ-phase formed. We indicate 1050 °C as the threshold temperature for maximum β-phase formation. That notwithstanding, the process must be optimized since previous studies have indicated that the presence of less than 10% γ-phase in the residue does not have a major adverse effect on lithium extraction efficiency. Should this study be valid, then we recommend 1000 °C thermal treatment for 60 min as an ideal decrepitation temperature for this concentrate considering the economics and quantity of γ-phase formed. However, if only a β-phase is required for industrial application, then a choice must be made at 1025 or 1050 °C, paying attention to the time required for the transformation, as well as the melting, which occurs at 1050 °C and its effect on downstream processing. In view of these uncertainties, we recommend further investigation between these temperatures which leads to good lithium extraction efficiency on this material.

Applying first-order kinetic models to the two processes provides a satisfactory fit to the experimental data and yields kinetic parameters and apparent activation energies of 655 and 731 kJmol^−1^, respectively for α- and γ-decay. SEM investigations reveal that, with increasing intensity of thermal treatments, spodumene grains undergo cracking, disintegration, followed by melting and agglomeration. 

## Figures and Tables

**Figure 1 materials-14-07423-f001:**
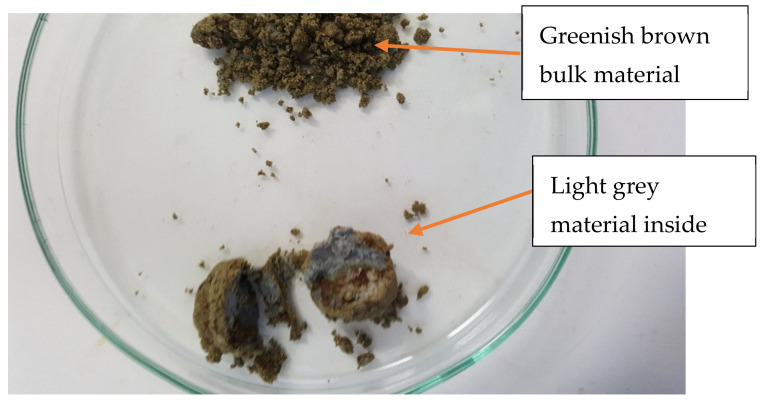
Optical image showing both the bulk and grey material contained in agglomerate particles.

**Figure 2 materials-14-07423-f002:**
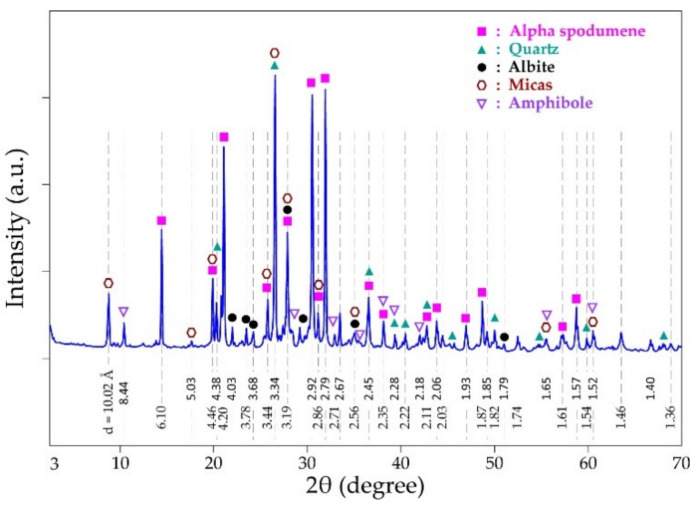
XRD patterns of spodumene concentrate.

**Figure 3 materials-14-07423-f003:**
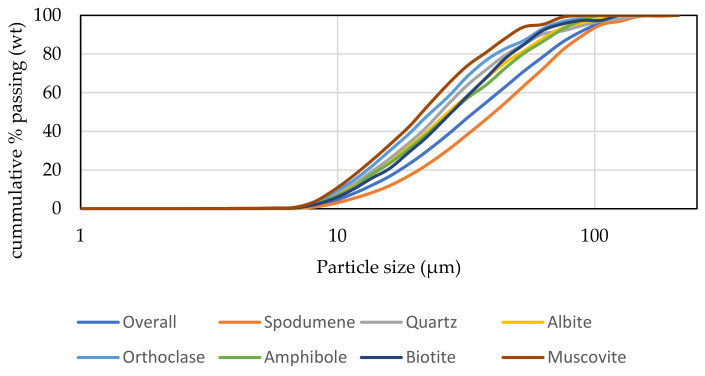
Mineral particle size determined by MLA.

**Figure 4 materials-14-07423-f004:**
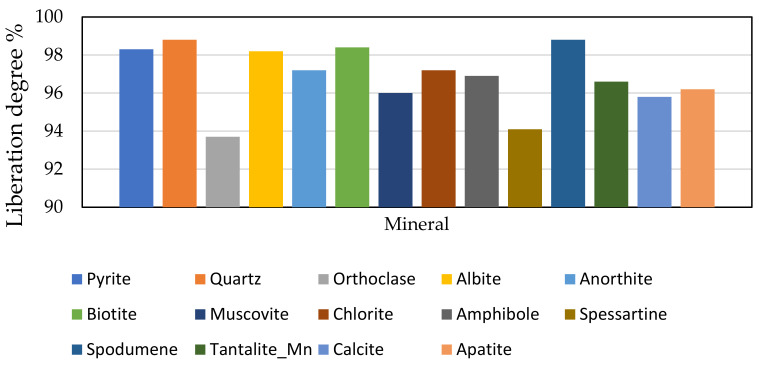
Degree of liberation of the main mineral phases.

**Figure 5 materials-14-07423-f005:**
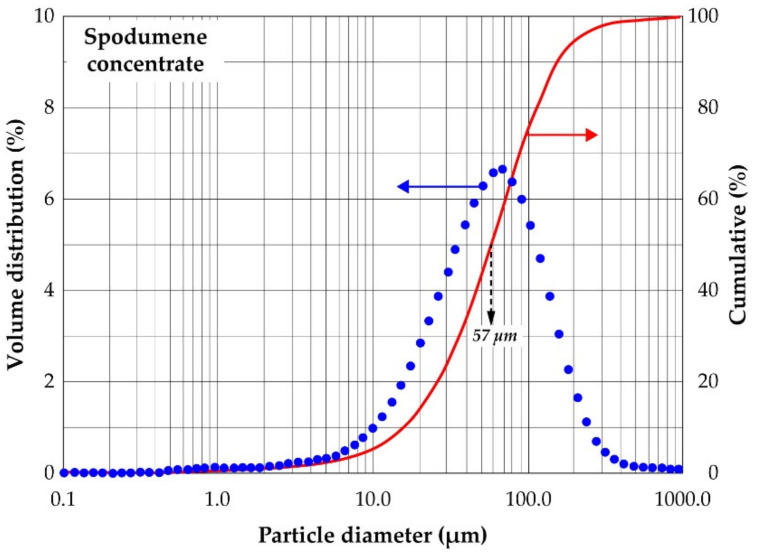
Volume distribution and cumulative passing of spodumene concentrate.

**Figure 6 materials-14-07423-f006:**
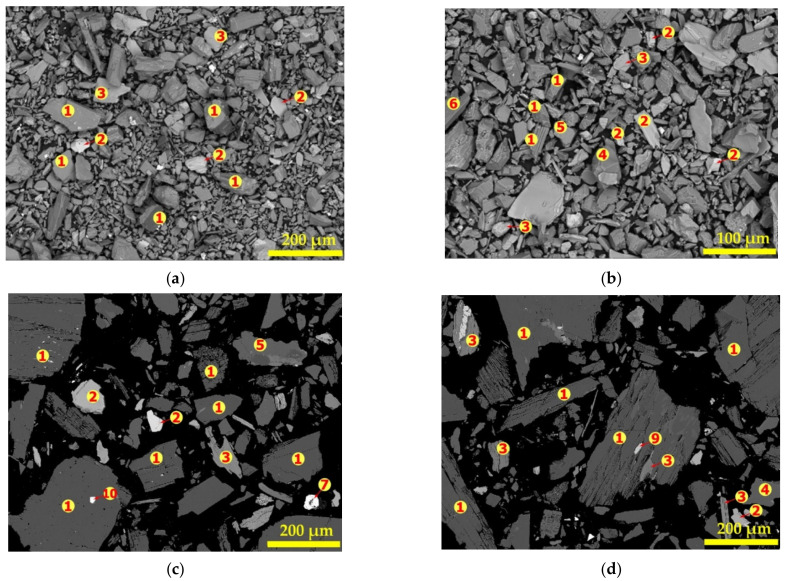
SEM of powdered (**a**,**b**) and polished samples (**c**,**d**) of concentrate. The numbers (1 to 7, 9, 10) indicate the locations of the elemental determination.

**Figure 7 materials-14-07423-f007:**
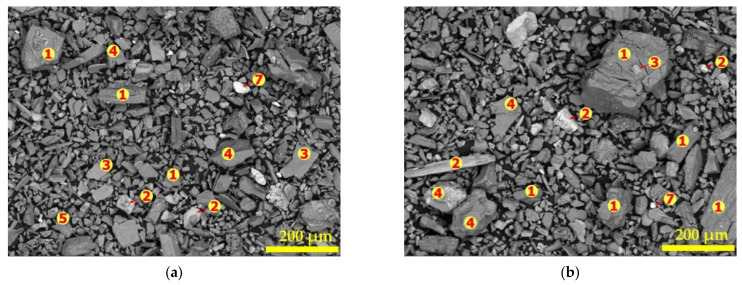
SEM and spot elemental determination of powdered (**a**–**d**) and polished sample (**e**–**h**) residues obtained after concentrate treatment in air at 900 °C (**a**,**e**), 950 °C (**b**,**f**), 1000 °C (**c**,**g**), 1050 °C (**d**,**h**) and melting and agglomeration occurring in square ”A” (**i**). The numbers (1 to 9) indicate the locations of the elemental determination.

**Figure 8 materials-14-07423-f008:**
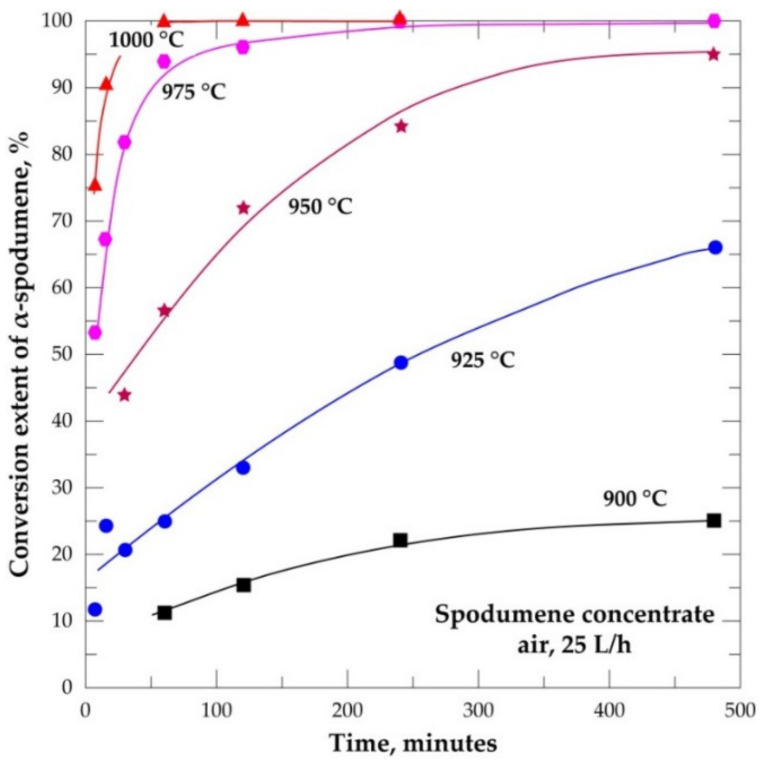
Conversion extent of α-spodumene into (β + γ) form during treatment of the concentrate between 900 and 1000 °C as a function of residence time.

**Figure 9 materials-14-07423-f009:**
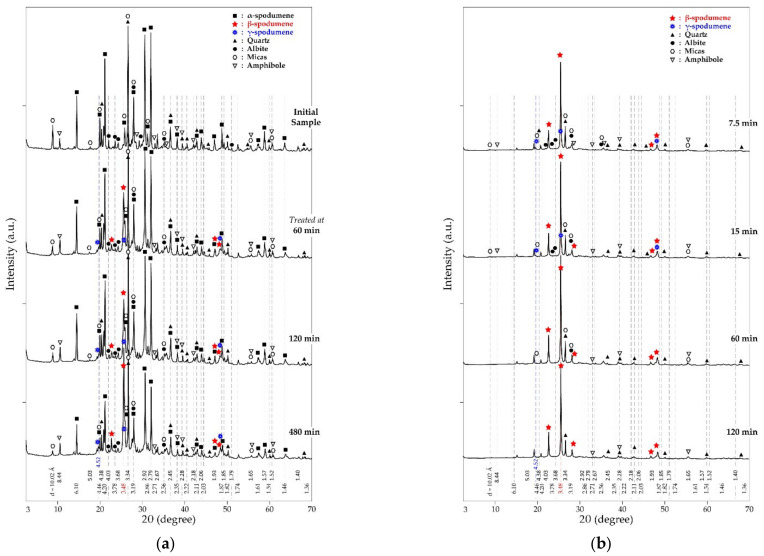
XRD patterns of residues obtained after concentrate treatment in air as a function of residence time at 900 °C (**a**) and 1050 °C (**b**).

**Figure 10 materials-14-07423-f010:**
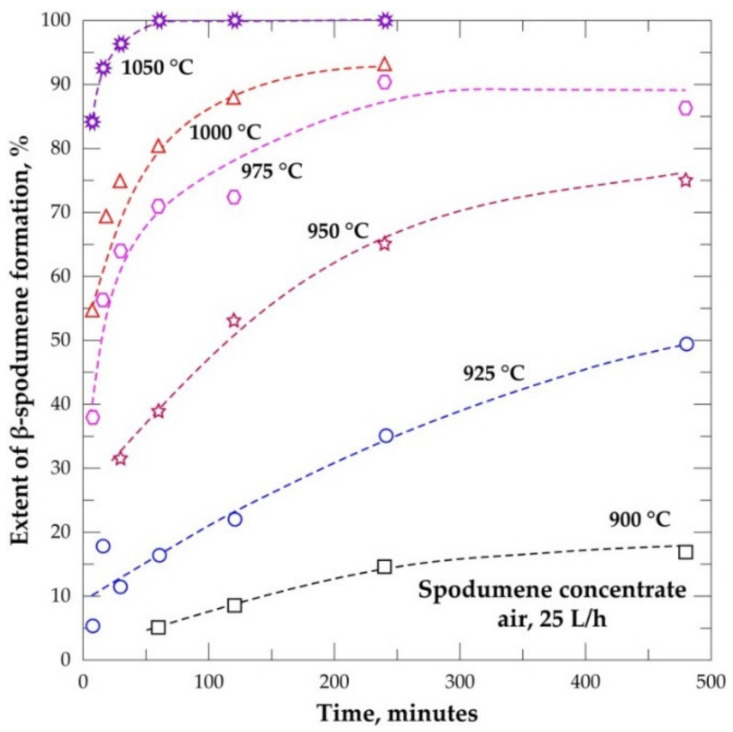
Spodumene formation during treatment of concentrate between 900 and 1050 °C as a function of residence time.

**Figure 11 materials-14-07423-f011:**
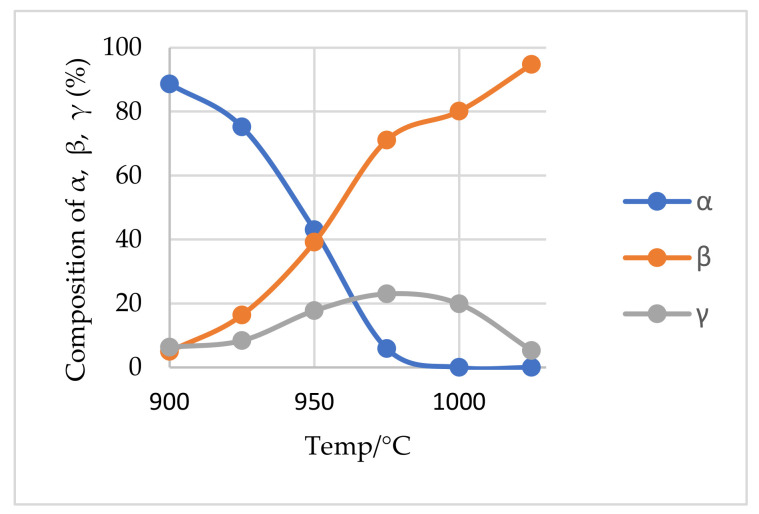
Percentage phase composition of α, β and γ as a function of temperature after 60 min treatment.

**Figure 12 materials-14-07423-f012:**
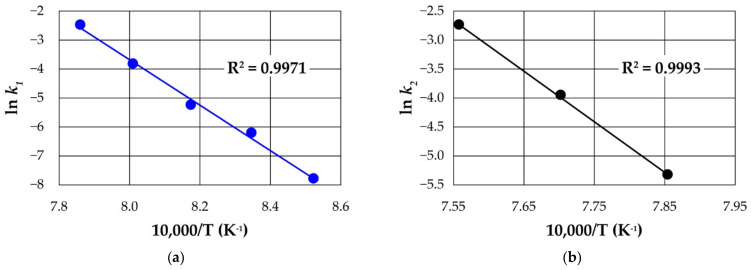
(**a**) ln $k_{1}$ against temperature (**b**) ln $$k_{2}$$ against temperature employing first-order kinetic model rate constants of decay plotted in Arrhenius form.

**Table 1 materials-14-07423-t001:** Elemental composition of concentrate determined by XRF and ICP-OES.

Major	Al_2_O_3_	CaO	Cr_2_O_3_	Fe_2_O_3_	K_2_O	MgO	MnO	Na_2_O	P_2_O_5_	SO_3_	SiO_2_	TiO_2_	Li	Cs	Rb	Ta	Nb	Sn
(wt.%)	20.79	1.72	0.03	4.29	1.26	1.30	0.32	1.14	0.33	0.15	61.31	0.10	2.14					
(ppm)														116	1033	202	180	140

**Table 2 materials-14-07423-t002:** MLA modal mineralogy.

Mineral	Formula	wt.%
Spodumene	LiAl(Si_2_O_6_)	60.21
Pyrite	FeS_2_	0.13
Quartz	SiO_2_	7.58
Orthoclase	KAlSi_3_O_8_	4
Albite	NaAlSi_3_O_8_	9.1
Anorthite	CaAl_2_Si_2_O_8_	1
Biotite	KMg_2.5_Fe^2+^0.5AlSi_3_O_10_(OH)_1.75_F_0.25_	3.32
Muscovite	KAl_3_Si_3_O_10_(OH)_1.9_F_0.1_	4.9
Chlorite	(Mg,Fe)_3_(Si,Al)_4_O_10_(OH)_2_·(Mg,Fe)_3_(OH)_6_	0.33
Amphibole	CaFeSi_2_O_6_	7.3
Spessartine	Mn_2_^+3^Al_2_(SiO_4_)_3_	0.61
Tantalite-(Mn)	MnTa_2_O_6_	0.12
Calcite	Ca(CO_3_)	0.47
Apatite	Ca_5_(PO_4_)(F,Cl,OH)	0.93

**Table 3 materials-14-07423-t003:** Calculated elemental assay from MLA.

Li	Al	Si	C	S	Fe	K	Na	Ta	P	Mn	Mg	Ca	Cl	F	H	O
1.96	11.58	29.47	0.06	0.07	2.0	1.34	0.76	0.09	0.17	0.22	0.5	1.97	0.02	0.09	0.04	49.39

**Table 4 materials-14-07423-t004:** Atomic percentage of mineral phases identified by SEM-EDS.

Elements	Spot “1”	Spot “2”	Spot ”3”	Spot “4”	Spot “5”	Spot “6”	Spot “7”	Spot “8”	Spot “9”	Spot “10”
O	64.7	60.9	54.4	66.7	61.9	61.6	60.1	67.3	56.6	67.8
Al	11.5	5.2	4.1		7.7	7.5	0.4			
Si	23.6	16.8	16.1	33.3	23.3	23.2	0.6			
Fe	0.2	6.2	2.2				38.9	1.9		
Mg		4.3	10.6				0.1			
Ca		4.8			0.2				20.2	
Mn		0.2						8.5		
Na		1.1			6.9	2.4				
K		0.5	3.9			5.2				
F			8.6						8.6	
P									14.6	
Nb								12.1		
Ta								10.3		0.4
N										6.2
Sn										25.6

**Table 5 materials-14-07423-t005:** Atomic percentage of some mineral phases identified by SEM-EDS at 1050 °C.

Elements	Spot “1”	Spot “2”	Spot “3, 5 and 6”	Spot “4”	Spot “7”	Spot “8”	Spot “9”
O	64.3		62.8	66.7	60.1	67.3	56.9
Al	11.4	3.4	6.6		0.4		
Si	23.1	19.5	24.3	33.3	0.6		
Fe	0.2	14.4	0.8		38.9	1.9	
Mg	0.2	3.9	0.3		0.1		
Ca	0.1	1.7	0.5				2.4
Mn	0.1	0.2				8.5	0.3
Na	0.6	0.9	2.0				
K	0.2	0.8	2.7				
Ti		0.3					
F							7.9
P							14.5
Nb						12.1	
Ta						10.3	

**Table 6 materials-14-07423-t006:** Reaction rate constants $k_{1}$ and $k_{2}$ estimated for α- and γ-decay, respectively.

T °C	900	925	975	1000	1025	1050
$k_{1}\;$(min^−1^)	0.0004	0.0019	0.0224	0.0054	0.0224	0.085
$k_{2}\;$(min^−1^)				0.0048	0.0192	0.065

## Data Availability

The data presented in this study are available upon request to the corresponding author.

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
