# Peer review of "Physico-Chemical Characteristics of Spodumene Concentrate and Its Thermal Transformations"

_materials, 2021, doi:10.3390/ma14237423_

Round 1

Reviewer 1 Report

Maybe because I am not familiar to this research filed, I do not find feel that the achievement of this paper is significant. Maybe this paper should emphasize the difference from the previous papers. Another problem is generalization of the results. This paper tested one spodumene concentrate sample. Is this sample able to be represent spodumene concentrate samples including the samples produced in different countries?

This paper should use Supplementary Material. The number of the figures in the present manuscript is too many, but more data should be provided, as mentioned below, to support the authors’ idea.

  1. The present figure number begins with 2.

  1. Line 91-92.

              Experimental method should provide sufficient information to reproduce the results. In this context, more detailed information on how the spodumene sample was provided is needed.

  1. Line 144

“MLA” should be “Mineral Liberation Analysis (MLA)”.

  1. Line 199

              I believe the authors collected many MLA data to summarize them in Fig. 5. A typical example of the MLA data should be provided as a Supplementary Material file.

  1. line 209

              The method used to obtain the volume distribution and cumulative passing of the concentrate is not provided.

  1. Line 211

              What is “d50” and “d80”?

  1. Table 4

              For example, there are eleven “1” in Figure 7(c) and (d). However, no standard deviations (errors) are provided for the data in Table 4. Are the values in Table 4 the averaged ones for plural particles or not?

              SEM-EDX cannot provide any data with four significant figures (e.g., 64.73).   

  1. Line 275

              What is the definition of “total disintegration”?

  1. Line 279, 287

              Figure 8h cannot indicate the agglomeration with impurities. Relevant MLA data should be shown in a Supplementary Material file.

              MLA data relevant with Line 287 should be shown in a Supplementary Material file.

  1. Table 5.

              The data at different temperature should be provided in a Supplementary Material file.

  1. Figure 9 and Figure 10.

              Figure 9 is a summary of the data in Figure 10. So, Figure 10 should be put before Figure 9, or Figure 10 should be put in a Supplementary Material file (the number of the figures of Figure 10 is too many).

  1. Line 436

              Which data did you use to calculate k1 and k2?

  1. Activation energy

              The equation used to calculate activation energy should be provided. The data used to calculate it should be specified. The fitting results (figures) should be provided in a Supplementary Material file.

  1. The second paragraph of the conclusion

              The contents of the second paragraph are not discussed in the Results and discussion section (they should be discussed).

Author Response

Thank you for your remarks to improve the paper. Find attached our responds and corrections (marked red in the manuscript) regarding your queries.

Reviewer 2 Report

Dear authors, I would like to make the following suggestions and comments:
1. Point 2 is very trivial, standard, widely known methods of analysis are described.
2. It is not clear why equations 3 and 4 were used for kinetic calculations. The obtained activation energy values look extremely high. In general, kinetic calculations are simplified. Including the last paragraph of the conclusions (for example, there is no data on the melting of materials in the text).
3. The article is very cumbersome and large, many of the described methods and results can be removed from the manuscript without losing the scientific essence.

Author Response

(The authors gave the same response as above.)

Round 2

Reviewer 2 Report

I am satisfied with the corrections and additions made in the text of the manuscript.

Author Response

Dear Sir,

Please, find attached the revised version of our manuscript and our reply to the Editor’s comments for the paper entitled ‘Physico-chemical Characteristics of Spodumene Concentrate and its Thermal Transformations’. We have taken into accounts each comment and the changes are in red color.

Editor comment 1

How the spodumene sample was provided is needed?

Authors’ reply 1

 Thank you for this question. We have added this information in the manuscript: “The concentrate produced as described above by Pilbara Minerals was used for this study”.

Editor comment 2

 "...BSE and X-ray signals produced from SEM..." it is not SEM to produce signals... "Mineral phase identification was achieved by X-ray analysis." which "X-ray analysis" do you mean?? XRD??? "The devise performs X-ray analysis on each grey level in the composite particles ..." Very confusing... which "devise"?? which "X-ray analysis" ??

Authors’ reply

In the scanning electron microscope, electrons bombarding the samples generate secondary electrons, backscattered electrons and x-rays which are detected. The x-ray detector is energy dispersive x-ray spectroscopy, not x-ray diffraction. The elemental composition of the minerals is measured by energy dispersive x-ray spectroscopy and the mineral phase is identified by matching with standard databases. The mineral phase mapping and quantification is then achieved by combining the map of phases identified with the backscattered electron images which resolve different mineral densities according to the signal brightness (greyscale). If not familiar with MLA, there is a good description of the approach via the following link (https://www.youtube.com/watch?v=oErWUVZUaMo). The MLA analysis for this research was carried out at the University of Queensland JKMRC Centre, where the MLA technology was originally developed.

Editor comment

All minerals in Table 2 are crystalline Peak shifts due to compositional variations of minerals are accounted for in the database. What XRD database has been used for identification of XRD patterns? Which software has been used for XRD identification? "Further research is however, recommended to ascertain these speculations." This further (better) use of XRD data must be performed in the present paper.

Authors’ reply

Mineral phases were estimated using the EVA© software coupled with the PDF2 database of the International Centre for Diffraction Data. Corrections have been effected in the text as “an operator decision was made to take into account only minerals which present at least two diffraction peaks (the main and secondary peaks). Thus, minerals with a very low concentration and showing only their main peak which were difficult to dissociate from the background noise could have been neglected. This could result to some trace minerals unidentified by XRD contrary to MLA and EDS point analysis (i.e. spessartine, tantalite and apatite)”.

Corrections have also been effected in conclusion as “Mineral phase identification by the analytical techniques employed in this study are consistent with each other for major minerals.

I hope this paper will be suitable for publication in Materials.

Best regards,

Prof. Alexandre Chagnes.

Director of the Industrial Partnerships at the Engineering School of Geology in Nancy

Director of the Labex Ressources21
